# Norethisterone Reduces Vaginal Bleeding Caused by Progesterone-Only Birth Control Pills

**DOI:** 10.3390/jcm11123389

**Published:** 2022-06-13

**Authors:** Naama Vilk Ayalon, Lior Segev, Abraham O. Samson, Simcha Yagel, Sarah M. Cohen, Tamar Green, Hila Hochler

**Affiliations:** 1Department of Obstetrics and Gynecology, Hadassah Medical Center, Faculty of Medicine, Hebrew University of Jerusalem, Jerusalem 9190401, Israel; naamaayalon@gmail.com (N.V.A.); simcha.yagel@gmail.com (S.Y.); sarahc362@gmail.com (S.M.C.); 2Science and Halacha Program, General History Department, Bar-Ilan University, Ramat-Gan 5290002, Israel; 3Puah Institute, Jerusalem 9547735, Israel; 4Faculty of Medicine, Bar-Ilan University, Safed 1311502, Israel; avraham.samson@biu.ac.il; 5Department of Obstetrics and Gynecology, Shaare Zedek Medical Center, Jerusalem 9103102, Israel; tamargreen1@gmail.com

**Keywords:** norethisterone acetate, POP, progesterone-only pills, vaginal bleeding, vaginal spotting, contraception

## Abstract

(1) Background: Progesterone-only pills (POP) are widely used contraceptives. About 40% of women taking these pills report vaginal bleeding/spotting; 25% find this a reason for cessation. To date, no effective remedy has been described. We aimed to examine the therapeutic approaches offered by health providers. (2) Methods: A prospective questionnaire-based study of women experiencing vaginal bleeding due to POP, comparing the effectiveness of prescribed therapies. Women were recruited through social networks, and subsequently divided into groups according to the treatment offered: (1) POP with norethisterone (*n* = 36); (2) double dose POP (*n* = 19); (3) single dose POP (continuing initial treatment, *n* = 57); and (4) different POP formula (*n* = 8). Women rated bleeding quantity and frequency at four intervals, at weeks 0, 2, 4, and 6. (3) Results: Women who added 5 mg norethisterone acetate reported a significant decrease in bleeding frequency compared to the other groups, observed after 2, 4, and 6 weeks (*p*-values 0.019, 0.002, and 0.002, respectively). Women also reported an overall decrease in combined bleeding quantity and frequency (*p*-values 0.028, 0.003, and 0.005, respectively). There was no difference in the rate of side effects among groups. (4) Conclusions: Adding 5 mg norethisterone acetate (Primolut-nor) to progesterone-only pills significantly reduces bleeding and spotting associated with POP contraception.

## 1. Introduction

Progesterone-only pills (POP) are widely used contraceptive agents, in particular by lactating women [1]. A substantial proportion of POP users—up to 40%—report spotting or irregular bleeding while on the pill [2]. Moreover, irregular bleeding and spotting are the main reason for discontinued POP use, and reportedly 25% of users stop using hormonal contraceptives due to irregular bleeding [3].

To resolve unscheduled bleeding associated with POP, several treatments have been proposed. Danielsson et al., reported a decrease in bleeding patterns when using antiprogestogen with POPs [4], although reanalysis of their data did not reveal any efficacy [5]. A very small study administering ethinylestradiol found no benefit [6]. Doubling the dose or switching the type of POPs have also been proposed; however, these approaches were also not shown to be effective [7]. Therefore, the Royal College of Obstetrics and Gynecology (RCOG) concludes that to date there are no effective treatments for this problem [8]. In clinical practice, some caregivers prescribe norethisterone acetate to reduce bleeding. To the best of our knowledge, this practice had not yet been tested.

In this study, we examine the effect of therapeutic approaches offered by health providers to treat menstrual irregularity associated with POP via an online questionnaire.

## 2. Materials and Methods

Women were recruited through social networks—Facebook groups aimed toward pregnant and lactating women, as well as the client mailing list of the Puah Institute, Jerusalem, Israel, which is a medical-halachic institute in Israel. Eligible participants were breastfeeding women up to six months after delivery, who experience irregular bleeding associated to POP. Exclusion criteria were switching to combined contraceptive pills, or failing to complete all questionnaires.

Recruited patients had been examined by their attending physician regarding medical history, to exclude other potential causes of their complaint and determine management.

Patients completed four online questionnaires. The first questionnaire (Appendix A) included questions about the socioeconomic status, medical background, the type of POP used, and questions characterizing the quantity and frequency of bleeding associated with the treatment offered to reduce irregular bleeding. This study did not include parity, and only measured the time from last delivery.

Women who met inclusion criteria were asked to complete three more follow-up questionnaires, two, four, and six weeks after the first one. In order to monitor bleeding patterns over time, and to improve the participant response rate, we used a biweekly questionnaire rather than the previously used daily one [9]. As such, we prepared a biweekly questionnaire based on existing recommendations for assessing bleeding under hormonal contraceptives [9]. These three follow-up questionnaires were identical (Appendix A) and asked women to rate the quantity and frequency of bleeding, as well as side effects of the treatment. The quantity of bleeding was measured using a five-point Likert scale, while the frequency was measured on a seven-point Likert scale, as detailed in Appendix A. Before statistical analysis, we translated both measurements to a scale of 10 [10].

This study was approved by the Ethics Committee of the Institutional Review Board (IRB permit 3.11.19) of Bar-Ilan University and was conducted between March 2019 and March 2020 in Israel.

Participants reported four treatments prescribed by their gynecologists to treat menstrual irregularity; they were categorized to four groups based on these responses:

(1) single dose POP (control, *n* = 57), (2) double dose POP (*n* = 19), (3) different POP formulation (*n* = 8), and (4) POP with norethisterone (*n* = 36). Women who received norethisterone were asked to stop treatment for 5 days, prior to adding norethisterone (Primolut-nor, 5 mg) in combination with POP.

### Statistical Analysis

To examine the effects of different treatments on bleeding patterns, we calculated the difference in the quantity and frequency of bleeding between the initial questionnaire and the three follow-up questionnaires, for each participant individually. In addition, we defined a variable combining frequency and quantity (with each variable weighted equally). Data analysis was carried out after translating the five- and seven-point Likert scale responses to a ten-point scale, according to orderly transformations [10].

We compared categorical variables using Chi-square or Fisher exact tests, according to the distribution of expected frequencies. According to the variable distribution and sample size, relationships between quantitative/ordinal variables were analyzed by Pearson or Spearman tests.

Differences between study groups were examined by t-test or by analysis of variance (ANOVA). If the ANOVA tests were statistically significant, we performed a multiple comparison applying the Benjamini–Hochberg procedure to control the false discovery rate (FDR) at a level of alpha = 0.05. Differences in ordinal variables were analyzed by Mann–Whitney or Kruskal–Wallis tests. Friedman test was performed to analyze the differences in bleeding rates between different questionnaires.

Statistical analysis was performed using the SPSS program, version 25.0. Statistical significance was determined as *p* < 0.05, and all tests were two-tailed. If the results of the parametric or asymmetric variance analysis tests were statistically significant, Bonferroni’s correction for multiple analyses was applied to control Type-I error.

## 3. Results

### 3.1. Demographics and Baseline Characteristics

The first questionnaire was answered by 174 patients. Excluded cases comprised 43 who did not complete all the questionnaires (37 failed to complete one questionnaire, 6 failed to complete two or more), five who reported no bleeding or spotting, and six who were using other forms of contraception concomitantly with POP.

Thus, a total of 120 women were included in the study. The vast majority of women (77%) reported use of Desogestrel 0.075 mg (Diamilla, Fominic, and Cerazette), while 23% reported use of Levonorgestrel 30 mcg (Microlut).

Women reported four different prescription treatments to treat menstrual irregularity and bleeding: (1) adding 5 mg norethisterone acetate (Primolut-Nor) after a five-day pause in taking the POP (*n* = 36), (2) doubling the POP dose (*n* = 19), (3) continuing using POP as before (*n* = 57), and (4) changing to another formulation of POP (*n* = 8).

The demographic and baseline characteristics of women taking each of these treatments were similar statistically, and their ages ranged from 20 to 44 years (Table 1). Notably, only eight women changed the POPs to another brand, which was a relatively small group compared to the other treatment groups, and disrupted the statistical comparison. As a result, this small group of women was omitted from our data analysis.

### 3.2. Treatment Response

Women who added 5 mg of norethisterone acetate to POP contraception reported a significant reduction in the frequency of bleeding compared to the other treatment groups after 2, 4, and 6 weeks of treatment (Table 2). In addition, these women reported more days of bleeding prior to treatment, emphasizing the results. This reduction in bleeding frequency was particularly pronounced in the comparison between women that added norethisterone acetate (group 1) and those who used a single dose POP (group 3, control).

In addition, adding norethisterone acetate resulted in a greater reduction in bleeding quantity, measured as the change in bleeding quantity between the first and the three subsequent questionnaires. Inconclusively, however, the comparison between different treatments was not statistically significant (Table 3).

A variable combining frequency and quantity of bleeding (with each variable weighted equally) also decreased when norethisterone acetate was prescribed, as compared to the other groups (Table 4). Notably, the reduction was statistically significant at each subsequent follow-up.

Table 5 shows side effects as reported by women in the fourth questionnaire, 6 weeks after starting the treatment. We found no significant difference in side effects throughout the study. Furthermore, participants were asked directly about the impact of the treatment on breastfeeding, and here also no significant difference was reported.

## 4. Discussion

Our study focused on resolving vaginal bleeding and spotting through the use of POP. Adding norethisterone acetate to POP after a five-day pause in taking the POP led to a reduction in bleeding, as compared to single and double POP doses. Women who used norethisterone acetate did not report any additional side effects.

We are aware of only three previous studies which examined treatments for vaginal bleeding caused by POP use. The largest and most recent of these [4] included 103 women and examined the addition of antiprogesterone (Org 31,710) vs. placebo once every 28 days. The authors found improvement in the bleeding profile; nonetheless, women continued to bleed every month, and the effectiveness of the treatment decreased over time. Notably, Org 31,710 is currently not marketed, and a reanalysis of the data showed no benefit [5]. The treatment, though effective, was based on a small cohort (20 women in the intervention group, both POP and IUD users), and required prolonged treatment (20 continuous days every month). Another study [6] examined 50 μg ethinylestradiol for 7 days vs. placebo. The cohort included only 12 women and the investigators found no benefit using this treatment. Indeed, the RCOG [7] indicates a lack of evidence for an effective treatment of bleeding in women using POP.

Few other studies have examined the effectiveness of different treatments on long term POP contraceptives, including injection of depot medroxyprogesterone acetate, subdermal etonogestrel, as well as levonorgestrel implants and intrauterine devices [5,11]. Treatments used were mifepristone, tranexamic acid, and NSAIDs, with conflicting results. We did not find studies that examined these treatments on POP users. In addition, these studies do not necessarily apply to POP, since other contraceptives were based on different progestin types and have different mechanisms of action.

As a potential limitation to this study, the mechanism of action of norethisterone acetate is not well understood. We hypothesize that its effect is based on its partial metabolism to ethinylestradiol in vivo. This metabolism was first reported in 1960 by Breuer and colleagues [12], who detected ethinylestradiol in the urine of women receiving norethisterone. Supporting evidence was provided by later in vitro studies in human placental microsomes, in homogenate of human liver, and in human hepatocytes [13,14,15]. Another study has examined the metabolism of norethisterone acetate and found that its metabolites bind to the human estrogen receptors [16].

Additionally, while the majority of women in this study received desogestrel, about one-quarter received levonorgestrel. We did not control for this variation in progesterone formulation.

Further studies in humans have attempted to estimate the conversion ratio of norethisterone acetate to ethinylestradiol. One study in 24 postmenopausal women who received norethisterone acetate 5 or 10 mg every morning found that every 5 mg of oral norethisterone acetate (one pill of Primolut-Nor) is equivalent to 28.1 ± 7.1 μg ethinylestradiol, and 10 mg norethisterone acetate is equivalent to 62.4 ± 18.6 μg ethinylestradiol [17]. In another study of 20 premenopausal women who received 10, 20, or 40 mg of norethisterone acetate daily for 7 days in the early follicular phase, the authors estimated the conversion ratio to be between 0.2% and 0.33%, which means that every 5 mg of oral norethisterone acetate is equivalent to 7.5 μg ethinylestradiol [18].

We hypothesize that the metabolite ethinylestradiol may have a beneficial effect on vaginal bleeding. It is well-known that exogenous estrogen may aid in tissue repair and stabilization of the endometrial lining [11]. It is thus plausible that norethindrone acetate works similarly in reducing bleeding. As mentioned above, the study that examined ethinylestradiol to treat vaginal bleeding due to POP showed no significant benefit [6]. Nevertheless, this might be due to the different dosage and the very small cohort of 12 women.

It is important to note that use of norethisterone acetate, which metabolizes into ethinylestradiol, warrants caution, especially in women with increased risk for thromboembolism or with other contraindications to exogenous estrogen [19].

Additionally, the conversion into ethinyl estradiol raises the question of prescribing combined oral contraception to breastfeeding women, which is a controversial issue [20,21]. We note that there are inconsistent data about the hormonal effects on milk quality and quantity, and about the passage of hormones to the infant. As a result, major organizations have adopted different recommendations: the WHO [22] and the RCOG [23] advise delaying use of combined oral contraceptives until 6 months postpartum for women who are primarily breastfeeding. In contrast, the CDC [24] allows the initiation of estrogen-containing contraceptive methods 6 weeks postpartum. Adding norethindrone to POP reportedly does not adversely affect the composition or the production of milk [25], and the passage of norethisterone acetate into nursing infants is expected to be minimal. It is likely that norethisterone acetate is also safe during lactation, although the manufacturer recommends cautionary use. In our study, we did not find more side effects on breastfeeding with norethisterone acetate compared to the other groups.

To the best of our knowledge, this is the first study to describe an effective treatment for unscheduled intermittent vaginal bleeding with POP. This is also the first report of norethisterone acetate to treat irregular bleeding during POP use. Our findings provide a useful strategy to reduce the bleeding profile of POP users. Since one of the main reasons for discontinuing POP is irregular bleeding and spotting, our findings may improve compliance and continuation rates, as well as patient satisfaction with POP.

Despite these encouraging results, this study has several limitations. First, this study is observational, and thus there might be a bias in the assigned groups and in the initial characteristics of women who added norethindrone acetate. Second, we used nonvalidated questionnaires.

To further confirm our results, additional studies may be required; for example, a prospective study to assess the effectiveness of norethindrone acetate compared to placebo and other treatments mentioned (e.g., mifepristone, tranexamic acid, and NSAIDs), as well as basic research to understand the exact mechanism of action of norethisterone acetate in reducing vaginal bleeding with POPs.

## 5. Conclusions

For women suffering from irregular vaginal bleeding associated with POP use, adding 5 mg norethisterone acetate (Primolut-Nor) after a five-day pause in taking the POP effectively reduces bleeding, without additional side effects.

## Figures and Tables

**Table 1 jcm-11-03389-t001:** Baseline demographic and obstetric characteristics of the study participants.

	Group 1 Norethisterone Acetate	Group 2 Doubling Dose	Group 3 No Change	Group 4 Switching Type	*p*-Value
N	36	19	57	8	
**POP TYPE**					
Desogestrel	29	14	42	8	0.18
Levonorgestrel	7	5	15	0	
**AGE**					
Mean ± SD	28 ± 5	26 ± 5	27 ± 5	28 ± 5	0.45
**EDUCATION**					
Non-academic	3	1	5	3	0.21
Academic	32	13	52	5	
**DELIVERY**					
Vaginal delivery	34	16	52	5	0.12
Cesarean section	2	3	5	3	
**BREASTFEEDING**					
Full	31	15	52	7	0.58
Partial	5	4	5	1	

**Table 2 jcm-11-03389-t002:** Comparison of bleeding frequency (mean ± SD).

		Group 1 Adding Norethisterone Acetate	Group 2 Doubling Dose	Group 3 No Change	*p*-Value *
Bleeding frequency at baseline (first questionnaire) on a scale of 10 ˣ		3.7 (±0.8)	3.2 (±1.3)	2.6 (±1.5)	0.001 ^1^
Difference from baseline to:	2 weeks	7.0 (±4.3)	4.6 (±4.5)	5.2 (±4.1)	0.019 ^2^
	4 weeks	8 (±3.78)	4.4 (±4.6)	4.8 (±4.8)	0.004 ^3^
	6 weeks	7.9 (±3.17)	5.4 (±4.5)	4.8 (±4.2)	0.002 ^4^

ˣ Data analysis was conducted by correcting the five-and seven-point Likert scale responses to a ten-point scale of infrequency. * *p* value by Kruskal–Wallis (First questionnaire) and by one-way ANOVA (Subsequent questionnaires). ^1^ A significant difference between groups 1 and 3 (*p* = 0.001). ^2^ A significant difference between groups 1 and 3 (*p* = 0.019). ^3^ A significant difference between groups 1 and 3 (*p* = 0.004), and also between groups 1 and 2 (*p* = 0.0021). ^4^ A significant difference between groups 1 and 3 (*p* = 0.002).

**Table 3 jcm-11-03389-t003:** Comparison of bleeding quantity (mean ± SD).

		Group 1 Adding Norethisterone Acetate	Group 2 Doubling the Dose	Group 3 No Change	*p*-Value *
Bleeding Quantity at baseline (questionnaire 1) on a scale of 10 ˣ		4.3 ± 1.7	1.6 ± 4.1	4.0 ± 1.6	0.703 ^1^
Difference from baseline to:	2 weeks	5.97 ± 3.39	3.75 ± 4.65	5.03 ± 3.25	0.294 ^2^
	4 weeks	6.57 ± 3.33	4.35 ± 4.99	5.19 ± 3.56	0.091 ^3^
	6 weeks	6.72 ± 2.86	5.39 ± 4.66	5.52 ± 3.29	0.246 ^4^

ˣ Data analysis was conducted by correcting the five-and seven-point Likert scale responses to a ten-point scale of inverse quantity. * *p* value by Kruskal–Wallis (First questionnaire) and by one-way ANOVA (Subsequent questionnaires). ^1^ Insignificant difference between groups 1 and 3 (*p* = 0.703). ^2^ Insignificant difference between groups 1 and 3 (*p* = 0.294). ^3^ Insignificant difference between groups 1 and 3 (*p* = 0.094). ^4^ Insignificant difference between groups 1 and 3 (*p* = 0.246).

**Table 4 jcm-11-03389-t004:** Comparison of combined frequency and quantity.

		Group 1 Adding Norethisterone Acetate	Group 2 Doubling the Dose	Group 3 No Change	*p*-Value *
Combined frequency and quantity at baseline (questionnaire 1) on a scale of 10 ˣ		8.17 ± 2.07	7.37 ± 2.11	6.60 ± 2.34	0.122 ^1^
Difference from baseline to:	2 weeks	6.49 ± 3.49	4.63 ± 3.61	4.52 ± 3.51	0.028 ^2^
	4 weeks	7.29 ± 3.25	4.40 ± 4.28	4.90 ± 3.39	0.003 ^3^
	6 weeks	7.33 ± 2.59	5.42 ± 3.97	5.09 ± 2.88	0.005 ^4^

ˣ Data analysis was conducted by correcting the five-and seven-point Likert scale responses to a ten-point scale of inverse quantity. * *p* value by Kruskal–Wallis (First questionnaire) and by one-way ANOVA (Subsequent questionnaires). ^1^ Significant difference between groups 1 and 3 (*p* = 0.028). ^2^ Significant difference between groups 1 and 3 (*p* = 0.003). ^3^ Significant difference between groups 1 and 3 (*p* = 0.005). ^4^ Insignificant difference between groups 1 and 3 (*p* = 0.246).

**Table 5 jcm-11-03389-t005:** Side effects at six weeks.

	Group 1 Norethisterone Acetate 5 mg	Group 2 Doubling Dose	Group 3 No Change	Group 4 Switching Type	*p*-Value
**No adverse effect**	26 (77%)	14 (82%)	39 (85%)	5 (83%)	0.198
**Decreased libido**	4 (12%)	0 (0%)	2 (4%)	0 (0%)	
**Headache/abdominal pain/dizziness**	0 (0%)	0 (0%)	1 (2%)	0 (0%)	
**Mood swings/depression/fatigue**	3 (9%)	2 (12%)	2 (4%)	1 (17%)	
**No change in breastfeeding**	25 (73%)	12 (70%)	43 (94%)	5 (83%)	0.083
**Decreased in breastfeeding**	8 (24%)	3 (18%)	3 (6%)	1 (17%)	
**Stopped breastfeeding**	1 (3%)	2 (11%)	0 (0%)	0 (0%)	

## Data Availability

Not applicable.

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
