# Peer review of "Norethisterone Reduces Vaginal Bleeding Caused by Progesterone-Only Birth Control Pills"

_jcm, 2022, doi:10.3390/jcm11123389_

Round 1
Reviewer 1 Report
Reviewer report
General comments
Congratulations to the authors for an interesting study.
However, some issues should be rectified before publication, such as the English editing of the text and Table 5, a better description of the volunteer group set, and a discussion section.
Minor comments
Introduction
Line 48- Please add the space before the reference [9].
Line 52- Please define how you examined the effects of different practices for the Readers within the description of the main goal in the study, more exactly through an online questionnaire.
Material and methods
Line 79- Please add the ethics committee approval number in the statement.
Section 3.1 lines 109-126- Please add the range of age from the volunteer group set indicating the clinical background as you reported in table 1. Just by table 1, the Readers only know the average age of each group and the number of each case, therefore a better description would beneficiate the authors for Readers’ comprehension of the results.
Table 5- line 179- Please rectify the percentage values of the table. It should be the number followed by the percentage values as “N (value%)” and not as “(%value) N”.
Discussion
Line 187- Please add the space before the reference [5]. Also, check the remaining references in the discussion section because there are other cases of the same type of mistake.
Line 207- Please write a proper sentence stating that there are several limitations of the present study. The authors only put the word limitations with a final remark point.
Lines 208-209- Please put “in vivo” in italics.
Author Response
Please find below a point by point reply to the comments. As such, we hope you find the paper suitable for publication.
General comments
Congratulations to the authors for an interesting study.
However, some issues should be rectified before publication, such as the English editing of the text and Table 5, a better description of the volunteer group set, and a discussion section.
>>>Thank you!
Minor comments
Introduction
Line 48- Please add the space before the reference [9].
>>> A space was added before reference [9]
Line 52- Please define how you examined the effects of different practices for the Readers within the description of the main goal in the study, more exactly through an online questionnaire.
>>>We thank the reviewer for this comment and have changed the sentence on line 52 to the following: “In this study, we examine the effect of therapeutic approaches offered by health providers to treat menstrual irregularity associated with POP via an online questionnaire.”
Material and methods
Line 79- Please add the ethics committee approval number in the statement.
>>>We thank the reviewer for this comment, and have added the ethics committee approval number, as following: “(IRB permit 3.11.19)”
Section 3.1 lines 109-126- Please add the range of age from the volunteer group set indicating the clinical background as you reported in table 1. Just by table 1, the Readers only know the average age of each group and the number of each case, therefore a better description would beneficiate the authors for Readers’ comprehension of the results.
>>>We thank the reviewer for this comment, and have added the range of age of the volunteer group, by table 1, on line 123, as following: “, and their age ranged from 20 to 44 years”.
Table 5- line 179- Please rectify the percentage values of the table. It should be the number followed by the percentage values as “N (value%)” and not as “(%value) N”.
>>>In accordance with the reviewers request, “(%value) N” was corrected to “N (value%)” in table 5.
Discussion
Line 187- Please add the space before the reference [5]. Also, check the remaining references in the discussion section because there are other cases of the same type of mistake.
>>>In accordance with the reviewers request, a space was added in front of reference 5, and where missing.
Line 207- Please write a proper sentence stating that there are several limitations of the present study. The authors only put the word limitations with a final remark point.
>>>We thank the reviewer for this comment, and have formulated a sentence on line 207 as following: “As a potential limitation to this study,…”
Lines 208-209- Please put “in vivo” in italics.
>>>In accordance with the reviewers request, “in vivo” has been italicized.
Reviewer 2 Report
Women who use progesterone-only (POP) pills as contraceptives report irregular vaginal bleeding/spotting. The authors recruited women mainly breast-feeding women, 6 months post-delivery. The women were divided into 4 groups depending on the type of POP offered to them. The authors found that women who used 5mg norethisterone along with progesterone-only (POP) pills experienced reduced bleeding and spotting than when using POP pills alone. The manuscript could be further improved by making these minor changes –
1. In page 1, line 35, the authors have used the word “contraindication”. Please define the word contraindication. It is unclear the way it is currently written.
2. Please rephrase the sentence “There are sparse ……..due to POP” – line 40, 41 on page 1. It is difficult to understand the way it is currently written.
3. The authors use the word “venotonic drug” on page 1, line 43. Please define the word “venotonic”.
4. The authors say they used a five-point Likert scale for quantity of bleeding and a seven-point Likert scale for frequency of bleeding. Please explain why the difference in scales for quantity and frequency of bleeding.
5. The authors used the word venotonic capillary protector on line 192. Please define the term venotonic capillary protector.
Author Response
Please find below a point by point reply to the comments. As such, we hope you find the paper suitable for publication.
Women who use progesterone-only (POP) pills as contraceptives report irregular vaginal bleeding/spotting. The authors recruited women mainly breast-feeding women, 6 months post-delivery. The women were divided into 4 groups depending on the type of POP offered to them. The authors found that women who used 5mg norethisterone along with progesterone-only (POP) pills experienced reduced bleeding and spotting than when using POP pills alone. The manuscript could be further improved by making these minor changes –
- In page 1, line 35, the authors have used the word “contraindication”. Please define the word contraindication. It is unclear the way it is currently written.
>>>We thank the reviewer for this comment, and have omitted the word contraindication entirely. The sentence is now as following: “Progesterone-only pills (POP) are widely used contraceptive agents, in particular by lactating women [1].”
- Please rephrase the sentence “There are sparse ……..due to POP” – line 40, 41 on page 1. It is difficult to understand the way it is currently written.
>>>We thank the reviewer for this comment and have rephrased the sentence as following: “To resolve unscheduled bleeding associated with POP, several treatments have been proposed.”
- The authors use the word “venotonic drug” on page 1, line 43. Please define the word “venotonic”.
>>>We thank the reviewer for this comment and have deleted this reference, as insufficient data is available about the venotonic compound used.
- The authors say they used a five-point Likert scale for quantity of bleeding and a seven-point Likert scale for frequency of bleeding. Please explain why the difference in scales for quantity and frequency of bleeding.
>>>We thank the reviewer for this comment. The difference in scales is based on different original perception, however the grades scales have been translated to a 10 point scale. This is reflected in the sentence: “Data analysis was carried out after translating the five- and seven-point Likert-scale responses to a ten-point scale, according to orderly transformations”
- The authors used the word venotonic capillary protector on line 192. Please define the term venotonic capillary protector.
>>>We thank the reviewer for this comment and have deleted this reference, as insufficient data is available about the venotonic compound used.
This manuscript is a resubmission of an earlier submission. The following is a list of the peer review reports and author responses from that submission.
Round 1
Reviewer 1 Report
Ayalon et al. report their findings from a prospective study conducted to evaluate the efficacy of norethisterone in combatting a common side-effect associated with the use of progesterone-only contraceptive pills, i.e., intermittent vaginal bleeding. They observed that significant reduction in intermittent vaginal bleeding can be achieved through the use of 5mg norethisterone by women who are using progesterone-only pills.
Major comments:
- It appears that for this study, the authors recruited women who were breastfeeding up to 6 months after delivery and who experience irregular vaginal bleeding. Did the authors take relevant personal or family history, such as cardiovascular disorders including thromboembolism, diabetes, reproductive history, gynecological conditions e.g. endometriosis, polycystic ovarian disorders, fibroids, into consideration? For all of the included women, did they experience intermittent vaginal bleeding only with the use of progesterone-only pills, or, did any of the women had pre-delivery history of spotting? Were any of these women on any other (non-contraceptive) medication?
- The authors mention that they recruited women for this study through social network. Further details on this should be provided.
- The authors divided the recruited women into 4 groups for addressing their intermittent vaginal bleeding into four groups, i.e., single dose POP, double dose POP, different POP formulation, POP with norethisterone. The authors should explain further their rationale behind this methodology. It is unclear whether the authors made any power calculations to determine the number of women that would be necessary for each group to make meaningful comparisons? The ‘different POP formulation’ group apparently comprised much lesser number of women (n = 8) than the other three groups (n = 57, n = 19, n = 36), and the authors eventually excluded this group from statistical analyses. Did the authors attempt to increase the sample size? Also, the majority of women in this study received desogestrel, and a smaller proportion received levonorgestrel. How did the authors correct for this bias?
- Information on the women that did not meet the inclusion criteria, i.e., the reasons for which they did not meet the inclusion criteria should be provided in a supplementary table.
- Did the authors gather information on the parity of these women? This should be included in Table 1.
Minor comments
- The authors need to pay attention to the formatting of the manuscript. The references provided do not have the numbers next to them, which makes it impossible to cross-check the references. Line 222 starts and ends with the word contraindications – it is unclear if this is mean to be a sub-heading.
Reviewer 2 Report
There are a lot of pitfalls in this study that preclude its publication: dessign, type of recruitment, creation of study groups, clear definition of inclusion and exclusion criteria, calculation of sample size,..
JCM better waits for better manuscripts